# Environmental fungi target thiol homeostasis to compete with *Mycobacterium tuberculosis*

**Neha Malhotra**[1¤a], **Sangmi Oh**[1], **Peter Finin**[1], **Jessica Medrano**[1¤b], **Jenna Andrews**[1¤c], **Michael Goodwin**[1], **Tovah E. Markowitz**[2], **Justin Lack**[2], **Helena I. M. Boshoff**[1], **Clifton Earl Barry, III**[1]*

**1** Tuberculosis Research Section, LCIM, NIAID, NIH, Bethesda, Maryland, United States of America, **2** Integrated Data Sciences Section, Research Technologies Branch, NIAID, NIH, Bethesda, Maryland, United States of America

¤a Current address: Center for Neural Circuits and Behavior, Department of Neurosciences, University of California San Diego, La Jolla, California, United States of America
¤b Current address: Department of Pediatrics, University of Pittsburgh Medical Center eeChildren's Hospital of Pittsburgh, University of Pittsburgh School of Medicine, Pittsburgh, Pennsylvania, United States of America
¤c Current address: Department of Microbial Pathogenesis, Yale University School of Medicine, New Haven, Connecticut, United States of America
* cbarry@niaid.nih.gov

**Data Availability Statement:** The fungal genome assemblies are deposited in the NCBI database under the BioProject PRJNA767328 with accession

## Abstract

Mycobacterial species in nature are found in abundance in sphagnum peat bogs where they compete for nutrients with a variety of microorganisms including fungi. We screened a collection of fungi isolated from sphagnum bogs by co-culture with *Mycobacterium tuberculosis* (*Mtb*) to look for inducible expression of antitubercular agents and identified 5 fungi that produced cidal antitubercular agents upon exposure to live *Mtb*. Whole genome sequencing of these fungi followed by fungal RNAseq after *Mtb* exposure allowed us to identify biosynthetic gene clusters induced by co-culture. Three of these fungi induced expression of patulin, one induced citrinin expression and one induced the production of nidulalin A. The biosynthetic gene clusters for patulin and citrinin have been previously described but the genes involved in nidulalin A production have not been described before. All 3 of these potent electrophiles react with thiols and treatment of *Mtb* cells with these agents followed by *Mtb* RNAseq showed that these natural products all induce profound thiol stress suggesting a rapid depletion of mycothiol. The induction of thiol-reactive mycotoxins through 3 different systems in response to exposure to *Mtb* suggests that fungi have identified this as a highly vulnerable target in a similar microenvironment to that of the caseous human lesion.

## Introduction

Tuberculosis remains a leading threat to global health and new chemotherapies to shorten the duration of treatment are urgently needed [1]. Current strategies to shorten the duration of treatment focus on attempting to understand bacterial metabolism in the microenvironment in which the bacilli persist and have centered on the caseous granuloma as one of the most difficult to sterilize Mycobacterium tuberculosis (*Mtb*) lesion types [2]. Within the caseous

numbers in Table 1. Fungal gene expression data was submitted to Gene Expression Omnibus (GEO)-NCBI database with GEO accession GSE271121 (F2); GSE271124 (F50); GSE271125 (F51); GSE271119 (C7); GSE271126 (F31) in mono- and co-culture conditions. Gene expression data for Mtb was submitted to GEO-NCBI database with accession number GSE255435.

**Funding:** This work was supported by the Division of Intramural Research NIAID, NIH (ZIA AI000693 to CEB). The funders had no role in study design, data collection and analysis, decision to publish, or preparation of the manuscript.

**Competing interests:** The authors have declared that no competing interests exist.

**Abbreviations:** ACP, acyl carrier protein; BGC, biosynthetic gene cluster; BUSCO, Benchmarking Universal Single-Copy Ortholog; CPM, count per million; FMO, FAD-dependent monooxygenase; ITS, Internal Transcribed Spacer; KS, ketoacyl synthase; MA, Magnitude-Amplitude; MIV, minimum inhibitory volume; Mtb, Mycobacterium tuberculosis; PBS, phosphate buffer saline; PDA, potato dextrose agar; PDB, potato-dextrose broth; PT, product template; TIC, total ion chromatogram.

granuloma nutrients are scarce, oxygen is limiting, and bacterial replication is slow. Pyrazinamide is perhaps the prototypical treatment-shortening drug, and its activity is highly pH dependent which has led to the suggestion that bacteria limiting treatment efficacy are likely within an acidic microenvironment [3]. There has been considerable effort to identify more "in vivo relevant" screening methods to identify potentially treatment-shortening leads and targets; however, these have largely been used to screen traditional compound libraries [4–7].

In considering what other sources of compounds might be used to identify targets that could result in treatment shortening, we considered environmental sources of competitors for slow-growing mycobacteria. In nature, *Mycobacteria* are found in a wide variety of reservoirs, largely, but not exclusively, aquatic in nature [8]. One important environmental reservoir for slow-growing mycobacteria is in the decomposing "gray-layer" underneath the zone of active growth of sphagnum peat bogs [9]. Mycobacteria are abundantly found in this layer reaching concentrations of up to $10^6$ cells per gram [9]. Microbiome sampling of sphagnum-associated bacteria have shown that 8% of the total community is composed of *Actinobacteria*, including *Mycobacteria* [10,11]. There is epidemiological evidence in Norway linking proximity to sphagnum peat bogs and the risk of developing leprosy, caused by the *Mtb*-related organism *Mycobacterium leprae* [12,13]. The characteristics of the "gray-layer" of a sphagnum peat bog has many similarities to the microenvironment within the caseous lesion of an *Mtb*-infected human lung in that it is nutrient-poor, hypoxic, and acidic.

Sphagnum bogs support a rich assemblage of microorganisms that have adapted to the acidic nutrient-poor environment created by these Bryophytes including many unique bacterial and fungal constituents [14]. We sampled sphagnum bogs throughout the Northeastern United States to identify fungal species occurring within this ecosystem that naturally compete with Mycobacteria to identify antitubercular compounds they may produce and what they have identified as the most vulnerable targets under these conditions.

## Material and methods

### Ethics statement

BSL-3 work was reviewed and approved by the NIH Division of Occupational Health and Safety and reviewed and approved by the NIH Dual Use Research Committee.

### Sampling sites, sphagnum collection, and fungal isolation

Sampling sites and peat collection: Sphagnum core samples were obtained from Cranberry Glades Botanical Area (Monongahela National Forest, WV) guided by Rosanna Springston (US Forest Service) on August 10, 2017; Hobart Bog (Moosehorn National Wildlife Refuge, ME, Special Use Permit MSH-18-015) on September 15, 2018; and Sunkhaze Meadows National Wildlife Refuge, ME, Special Use Permit MSH-18-015) on September 16–18, 2018 (S7 Table). Soil was collected using a Russian Peat Borer (Aquatic Research Instruments, Hope, ID) from various depths ranging from the top layer to about 50 inches deep under the surface. About 50 g of soil mass was carefully collected in 50 ml falcon tubes following sterile procedures and stored at 4°C until processed.

Fungi isolation: 5 g of sphagnum core mass was suspended in 30 ml 0.1 M phosphate buffer containing 0.01% Tween-80, pH 4.0 in a 50 ml falcon tube with 4 mm glass beads (about 5 ml). The samples were vortexed vigorously for 2 min and allowed to settle for 30 min. The top layer of the soil suspension was plated at different dilutions on different solid agar media; potato dextrose agar (PDA, pH 5.4), pectin enriched minimal media [15] (pH 4.0) and sphagnum enriched media (pH 4.0) with 100 μg/ml chloramphenicol, and 50 μg/ml gentamicin sulfate. Plates were incubated for 5 to 15 days at 25°C and visible fungal colonies were subcultured by

transferring spores onto fresh plates. Pure spores were finally collected using phosphate buffer with 0.1% Tween-80 (pH 6.0). Spores were counted using an automated cell counter (Nexcelom Bioscience, Lawrence, Massachusetts) and the suspensions were stored in 2 ml aliquots in 96-well deep-well plates at −80°C.

## Bacterial strains and culture media

*Mtb* H37Rv was grown in Middlebrook 7H9 medium (Difco Laboratories) supplemented with 0.5% Albumin, 0.2% D-Glucose, 0.2% Glycerol, 0.085% NaCl, 0.05% Tween-80 (ADGNTw) or 7H11 media supplemented with ADGN and 0.006% Oleic acid (OADGN). The reporter strain pOLYG-mScarlet *Mtb* H37Rv has been previously described [16]. Cells expressing pPMF62a-3054p (see below) as well as Mrx1-roGFP2 reporter plasmids [17] were grown in 7H9+-ADGNTw containing 50 μg/ml hygromycin. To prepare aliquots of *Mtb* for fungal induction 2 L detergent and BSA-free 7H9 media (supplemented with 4 g/L Glucose, 0.81 g/L NaCl, and 0.3 g/L Casitone) was inoculated with 200 ml of wild-type *Mtb* H37Rv (OD$_{600}$ 1.0) in roller bottles and incubated at 37°C for 2 weeks. The cells were harvested by centrifuging at 4,000 rpm for 30 min and were washed with 200 ml phosphate buffer saline (PBS). The cells were re-washed as above and finally resuspended in 200 ml PBS to form 10× concentrate (X refers to the density of a fully saturated culture). The cells were homogenized using glass beads (4 mm, 20 ml) by vortexing for 5 min. This 10× cell suspension was then aliquoted into cryovials and frozen at −80°C for further use. Concentrated *Mtb* stocks were serially diluted in phosphate buffer (pH 6.0) and homogenized by using glass beads (4 mm, 5 ml) followed by vortexing for at least 2 min. *Enterococcus faecium*, *Staphylococcus aureus*, *Klebsiella pneumoniae*, *Acinetobacter baumannii*, *Pseudomonas aeruginosa*, and *Enterobacter* species were cultured in brain–heart infusion broth (Difco Laboratories). These bacteria were recent clinical isolates obtained from Dr. Adrian Zelazny in the NIH Clinical Microbiology laboratory.

## Co-cultivation based fungal hits screening and growth inhibition assays

Preliminary screening of the induced fungal hits was performed in either soft-solid PDA containing 0.75% agar in 96-well trans-well plates (Corning Inc.) and/or potato-dextrose broth (PDB) in 12- or 24-well multi-dishes. Within each well of the insert plate, 100 to 1,000 fungal spores from frozen stocks were added on the surface of soft-solid PDA. Each well of the donor plate was immersed in 200 μl of phosphate buffer (pH 6.0) within the receiver plate. The plates were incubated at 28°C for 24 h followed by the addition of different concentrations of *Mtb* inducer. The plates were further incubated for 9 to 15 days at 28°C to 32°C. Liquid co-cultivation was performed in either 12- or 24-well plates in PDB media (1 to 1.2 ml/well) containing approximately 1,000 spores grown for 24 h at 28°C before adding *Mtb* inducer and incubating at 28°C to 32°C for 9 to 15 days with shaking (45 rpm). The contents of the receiver plate and the liquid cultures (mono and co-cultures) were filtered through 0.2 μm 96-well filter plates (MilliporeSigma, Burlington, Massachusetts) and 2-fold serial dilutions were done in 96-well clear-bottom black plates (Corning Inc.) in 7H9 media supplemented with ADGNTw and hygromycin (50 μg/ml); 10$^4$ cells of pOLYG mScarlet *Mtb* H37Rv were added, and the plates were incubated at 37°C. Growth of the reporter strain was monitored on Day 7 and Day 14 using a Clariostar (BMG Labtech, Cary, North Carolina) plate reader at Em: 610 nm (Ex: 570 nm). The volume of the filtrate at which there was 90% to 100% growth inhibition was defined as the "Minimum Inhibitory Volume" or "MIV." The cidality assay for the *Mtb* H37Rv strain was performed in 24-well plates after 7 days of exposure of cells to the induced fungal culture filtrates. *Mtb* H37Rv culture in 7H9-ADGNTw media was grown to an OD$_{600}$ of 0.6 and was filtered through a 5 μm filter to remove cell clumps. Cells were diluted to OD$_{600}$ 0.02 to 0.03

and appropriate dilutions plated on 7H11-OADGN (ADGN supplemented with oleic acid) media to enumerate Day 0 inoculum. The remaining culture was dispensed in 24-well tissue culture plates (0.5 ml/well) containing concentrations of fungal filtrates corresponding to 2× and 10× MIV and incubated at 37°C, 40 rpm for 7 days. Treated cells were washed and plated on 7H11-OADGN media for CFU enumeration. The assay was performed for 2 biological replicates. CFU was counted after incubation of the plates at 37°C for at least 3 to 4 weeks and bactericidal activity was determined in terms of decrease in $\log_{10}$ CFU/ml.

## Fungal genomic DNA isolation

Fungal cultures for the positively induced hits were obtained by inoculating about 50,000 spores in 50 ml PDB media (100 μg/ml chloramphenicol and 50 μg/ml gentamicin sulfate) in 250 ml roller bottles (28°C, 5 to 7 days, 75 rpm). Genomic DNA isolation was performed using the protocol described elsewhere [18] with some modifications. Mycelia were harvested and washed with phosphate buffer (pH 6.0) 3 times by centrifuging at 4,000 rpm, 10 min, RT followed by vacuum filtering to remove excess media components. The mycelia were then ground to fine powder with liquid $N_2$ by mortar-pestle and stored at −80°C until further use. Cell lysis was performed in tubes with ground mycelia by adding Zr/Si (0.1 mm, 375 μl) beads in 800 μl extraction buffer (0.1 M Tris-HCl (pH 8.0); 10 mm EDTA (pH 8.0); 2.5 N NaCl; 3.5% CTAB) and 20 mg/ml proteinase K (180 μl). The lysate was mixed and incubated at 65°C for 45 min. After centrifuging (13,000 rpm, 15 min, RT), an equal volume of phenol-chloroform-isoamyl alcohol (25:24:1) was added to the supernatant. Samples were mixed by inverting for 15 s and kept at RT for 1 h followed by centrifuging as above. An equal volume of chloroform-isoamyl alcohol was added to the aqueous phase and mixed. After centrifugation, an equal volume of ice-cold isopropanol was added to the aqueous phase. Samples were incubated at −20°C for at least 2 h to precipitate the DNA followed by centrifugation as above. The supernatant was completely decanted, and the DNA pellet was washed with ice-cold 70% ethanol (1 ml, 3 times) by centrifugation (13,000 g, 1 min, RT). The DNA pellets were air-dried overnight and dissolved in 30 to 40 μl molecular biology grade DNase free water and stored at −80°C. To ensure no contamination of fungal genomic DNA with bacterial DNA, PCR amplification for 16s rRNA was carried out using Q5 high fidelity polymerase and primers 16s-27F and 16s-805R (S1 Table).

## Internal transcribed spacer analysis and whole genome sequencing

Preliminary identification of the fungal hits with induced activity was carried by PCR amplifying the Internal Transcribed Spacer (ITS) region using the ITS5 and ITS4 primers [19] (S1 Table). The ITS PCR products were extracted from a 1% agarose gel and were sequenced using ITS5 primers. NCBI BLASTn was used to identify the closest match using the ITS database.

Whole genome sequencing of the fungal hits was done utilizing Large Nanopore Combo and Hybrid Assembly service from SeqCenter, PA. For short read sequencing, sample libraries were prepared using the Illumina DNA Prep kit and IDT 10 bp UDI indices, and sequenced on an Illumina NextSeq 2000, producing 2 × 151 bp reads. Demultiplexing, quality control, and adapter trimming was performed with bcl-convert (v3.9.3). For long read sequencing sample libraries were prepared using Oxford Nanopore Technologies (ONT) LigationSequencing Kit (SQK-LSK109) with NEBNext Companion Module (E7180L) in addition to Native Barcode Kits (EXP-NBD104, EXP-NBD114) to manufacturer's specifications. All samples were run on Nanopore R9.4.1 flow cells and MinION Mk1B device. Post-sequencing, Guppy (v5.0.16) was used for high accuracy base calling (HAC) and demultiplexing [20]. Quality control and adapter trimming was performed with bcl-convert [21] and porechop (https://github.com/rrwick/Porechop) for Illumina and ONT sequencing, respectively. Long read assembly

**Table 1. Summary of assembly metrics for the fungal genomes identified in the study.**

| Fungal Id | # Contigs | Total Length (Mb) | GC (%) | N50 (Mb) | N75 (Mb) | BUSCO Score (%) | Nearest Neighbor (ITS BLAST) | Accession |
|---|---|---|---|---|---|---|---|---|
| F2 | 219 | 32.92 | 49.74 | 0.56 | 0.30 | 80.8 | *Penicillium rolfsii* | JAYXSC000000000 |
| F50 | 85 | 33.04 | 49.57 | 2.68 | 1.59 | 82.4 | *Penicillium rolfsii* | JAYXSA000000000 |
| F51 | 78 | 32.98 | 49.60 | 2.76 | 1.89 | 83.9 | *Penicillium rolfsii* | JAYXSB000000000 |
| C7 | 163 | 32.29 | 47.92 | 2.38 | 0.93 | 80.0 | *Penicillium lividum* | JAYXSD000000000 |
| F31 | 61 | 34.80 | 45.69 | 2.59 | 1.66 | 74.1 | *Talaromyces stollii* | JAYXSF000000000 |

with ONT reads was performed with flye version 2.8 [22]. The long read assembly was polished with pilon [23]. To reduce erroneous assembly artifacts caused by low-quality nanopore reads, long read contigs with an average short read coverage of 15× or less were removed from the assembly. Assembly statistics were recorded with QUAST [24]. Assembly annotation was performed with funannotate [25]. BUSCO (Benchmarking Universal Single-Copy Ortholog) version 4.1.3 was used to assess genome completeness while N50 and N75 were calculated to determine genome contiguity. Phylogenetic analysis of the fungal isolates was done utilizing the online version of CVTree 3.0. Genome sequences (.faa) of the closest neighbors based on the BLAST search of Internal Transcribed Spacer (ITS), β-tubulin and calmodulin regions of the induced fungal isolates were used to perform the phylogenetic assessments. The fungal genome assemblies are deposited in the NCBI database under the BioProject PRJNA767328 with accession numbers in Table 1.

## Fungal RNAseq and differential gene expression

Fungal biomass from mono- and co-cultures of the induced potential hits was obtained by inoculating about 300,000 spores in 4 ml PDB media in 6-well plates (Corning Inc.). The cultures were grown at 28°C for 1 day. For co-cultures, 450 μl of 1× (for F2/F50/F51 fungi), 0.1× (for C7 fungus), and 0.001× (for F31 fungus) dilutions of *Mtb* inducer stock resuspended in phosphate buffer (pH 6.0) were added to fungal cultures on the next day and the cultures were incubated at 32°C at 45 rpm. All culture sets were carried out in at least 2 biological replicates. For monocultures an equivalent volume of phosphate buffer (pH 6.0) was added. Mycelia were harvested on days 6, 9, and 12 by centrifuging at 4,000 rpm for 20 min followed by a brief washing with DEPC-treated water (10 ml, 4,000 rpm, 20 min). The harvested fungal biomass was ground to fine powder in liquid nitrogen with a mortar-pestle, which was then transferred to 2 to 3 ml TRIzol Reagent (Invitrogen, Waltham, Massachusetts) to form a slurry and finally thawed. The samples were pipetted several times to ensure complete homogenization and were split into 2 to 3 RNase-free microcentrifuge tubes (1 ml each) and incubated for 5 min at RT. Chloroform (300 μl) was added to each tube and these were shaken vigorously. The samples were centrifuged at 13,000 × g at 4°C for 15 min. To the upper aqueous layer, an equal volume of 70% ethanol was added to precipitate the RNA, which was then vortexed and loaded onto spin cartridge columns provided by PureLink RNA isolation kit (Invitrogen, Waltham, Massachusetts). Further purification steps were done according to the manufacturer's guidelines with an on-column DNase I treatment (Zymo Research, Irvine, California). The RNA was eluted in 40 μl DEPC-treated RNase-free water and stored at −80°C. RNA concentration and purity were analyzed using Nanodrop spectrophotometer (Thermo Fisher Scientific, Waltham, Massachusetts) followed by visual assessments using an Agilent Bioanalyzer 2100 (Agilent, Santa Clara, California) and denaturing formaldehyde agarose gel electrophoresis (1.3% agarose, formaldehyde in 1× MOPS buffer). Total RNA (at least in 2 biological replicates) from

mono- and co-culture conditions were used for mRNA sequencing following ribosomal RNA depletion and cDNA Library prep. Likewise, for expression analysis of citrinin in C7-*Mtb* and C7+*Mtb* cultures, total RNA was extracted from both conditions in at least 2 biological replicates. cDNA preparation and real-time quantification for the targeted regions of citrinin gene cluster (C7_116, C7_118, and C7_120; primers listed in S1 Table) was performed using One step Luna Universal RT-qPCR kit (New England Biolabs, Ipswich, Massachusetts) using an Applied Biosystems QuantStudio 3 (Thermo Fisher Scientific, Waltham, Massachusetts).

Libraries for mRNA sequencing were constructed from total RNA sample using the Zymo-Seq RiboFree Total RNA Library Prep Kit (Zymo Research, Irvine, California) according to the manufacturer's instructions. Libraries were amplified to incorporate full-length adapters. RNAseq libraries were sequenced on an Illumina NovaSeq 6000 to a sequencing depth of at least 30 million read pairs (150 bp paired-end sequencing) per sample. The Zymo Research RNA-Seq pipeline was originally adapted from nf-core/rnaseq pipeline v1.4.2 (https://github.com/nf-core/rnaseq) [26]. Briefly, adapter and low-quality sequences were trimmed from raw reads using Trim Galore! v0.6.6 (https://www.bioinformatics.babraham.ac.uk/projects/trim_galore). Trimmed reads were aligned to the reference genome using STAR v2.6.1d (https://github.com/alexdobin/STAR) [27]. BAM file filtering and indexing was carried out using SAMtools v1.9 (https://github.com/samtools/samtools) [28]. RNAseq library quality control was implemented using RSeQC v4.0.0 (http://rseqc.sourceforge.net/) [29] and QualiMap v2.2.2-dev (http://qualimap.conesalab.org/) [30]. Duplicate reads were marked using Picard tools v2.23.9 (http://broadinstitute.github.io/picard/). Reads overlapping with exons were assigned to genes using featureCounts v2.0.1 (http://bioinf.wehi.edu.au/featureCounts/) [31]. Differential gene expression analysis was completed using DESeq2 v1.28.0. Differentially expressed sequences were assembled using the annotation.gbk files for their respective genomes obtained from SeqCenter. Fungal differential expression was depicted as Magnitude-Amplitude (MA) plot representing log ratio fold change ($\log_2$FC) as "M" on y-axis while mean average of gene expression in terms of DESeq2 normalized mean read counts is shown on x-axis using DESeq2 v1.28.0 (https://bioconductor.org/packages/DESeq2/) [32]. Genes with negative $\log_2$FC have higher expression in co-culture conditions, while positive $\log_2$FC represents the up-regulation in monoculture conditions. Nucleotide and protein sequences for contiguously placed induced genes unique to co-culture conditions were extracted using Geneious Prime [33]. Protein homology was assessed using BLASTx against experimental clustered Nr database (https://blast.ncbi.nlm.nih.gov/Blast.cgi). Protein domain and functional annotation was also determined using InterPro (https://www.ebi.ac.uk/interpro/search/sequence/) and UniProt (https://www.uniprot.org/). Sequence alignments were done by CLUSTALO (https://www.ebi.ac.uk/Tools/msa/clustalo/) and phylogenetic analysis by MEGAX [34] and Geneious prime software using the Maximum Likelihood algorithm as well Neighbor-joining methods. BGC annotation and characterization in the fungal genomes was done using antiSMASH fungal version 7.0 [35]. Gene expression data was submitted to Gene Expression Omnibus (GEO)-NCBI database with GEO accession GSE271121 (F2); GSE271124 (F50); GSE271125 (F51); GSE271119 (C7); GSE271126 (F31) in mono- and co-culture conditions.

### *Mtb* RNA isolation and differential gene expression

*Mtb* (OD$_{600}$ 0.2, 10 ml) cells were exposed to concentrations equivalent to 1× and 10× MIV of induced fungal filtrates (F2+*Mtb* and F31+*Mtb*). Cells were harvested after 6 h and 12 h exposure and were washed twice with 7H9-ADGNTw broth by centrifuging (4,000 rpm, 4°C, 10 min). Cells were resuspended in 0.9 ml TRIzol and were lysed by bead beating using 0.1 mm Zr/Si beads (6000, 2 to 3 cycles, 30 s; 5 min on ice between pulses). The resulting lysate was

centrifuged (13,000 rpm, 1 min, 4˚C) to remove the cell debris and 0.3 ml chloroform (Sigma) was added to the supernatant. Tubes were thoroughly shaken, allowed to sit for 5 min and centrifuged (13,000 rpm, 15 min, 4˚C) to separate the upper aqueous layer. Total RNA was precipitated using equal volume of 70% ethanol and vortexed for 30 s. The solution containing RNA was loaded onto spin cartridge columns and the RNA was eluted according to the manufacturer's guidelines (PureLink, Invitrogen, Waltham, Massachusetts). Total RNA was stored in aliquots at −80˚C. Total *Mtb* RNA was used for cDNA library prep and mRNA sequencing in 3 biological replicates for 4 different conditions; 6 h, 1×; 12 h, 1×; 6 h, 10× and 12 h, 10×. mRNA sequence obtained from untreated *Mtb* was used as a control for differential gene expression analyses. About 100 ng of Total RNA was used to produce mRNAseq samples using the Illumina TruSeq Stranded Total RNA Gold kit and sequenced as 151 bp paired-end libraries on an Illumina NextSeq 2000 P2. cDNA prep and mRNA sequencing were done at the sequencing facility of Fredrick National Laboratory for Cancer Research (FNLCR, Fredrick, Maryland, USA). The data was processed using the RNA-seek workflow v1.8.0 (https://doi.org/10.5281/zenodo.5223025) on the NIH HPC Biowulf cluster (http://hpc.nih.gov) with minor adaptations. References were created using the–small-genome build command. Reads were then trimmed using cutadapt v1.18 and aligned to the NC_000962.3 reference genome using STAR v2.7.6a in 2-pass basic mode with the following adaptations to handle the small, non-intronic genome: ALIGNINTRONMAX = 1, ALIGNINTRONMIN = 0, and ALIGNMATESGAPMAX = 5,000 [27]. Expression levels were quantified with RSEM v1.3.0 [36]. For differential analyses, genes with a read count per million (CPM) less than 0.5 in at least 3 samples were filtered before running limma 3.52.4 [37]. Genes were identified as significant with an FDR of 0.1 and an absolute fold-change of 2. Venn diagrams were created with ggvenn 0.1.10 (https://www.rdocumentation.org/packages/ggvenn/versions/0.1.10). Log-fold change heatmaps were created using ComplexHeatmap 2.12.1 with circlize 0.4.15 and RcolorBrewer 1.1–3 to control the colors. Cluster networks were performed using STRING db (https://string-db.org/) and functional annotation of the genes was according to Mycobrowser (https://mycobrowser.epfl.ch/) and TB Database (http://tbdb.bu.edu/). Differential expressions at different treatment conditions are represented by MA plot, where x-axis represents the log CPM while y-axis is $\log_2$FC using Glimma v2.6.0 [38]. Gene expression data for *Mtb* was submitted to GEO-NCBI database with accession number GSE255435.

## Reactivity-based screening by DTT adduct assay

DTT adduct assay was performed as previously described by treating 20 µl of induced F31 +*Mtb* media filtrate with 20 µl of 1 M dithiothreitol (DTT) in basic conditions (diisopropylethylamine 2.5 µl/ml, final concentration 20 mM) in methanol [39]. Control samples were treated similarly but without DTT. Following 16 h of incubation protected from light, metabolite analysis was done by LC-MS/MS by scanning both positive and negative mode ESI from m/z 100 to 1,000 for F31+*Mtb* samples with and without DTT. Chromatograms were compared by examining the -DTT samples for peaks absent in the +DTT samples and vice versa with signal abundance of more than 100,000.

## Extraction, fractionation, and identification of active metabolite in F31 +*Mtb* and C7+*Mtb* culture filtrate

Fungal filtrates from a 15-day co-cultivation of F31+*Mtb* and C7+*Mtb* in 6-well plates were filtered twice and pooled separately to obtain about 400 ml total filtrate volume from each. Filtrates obtained were mixed with 250 ml ethyl acetate (EtOAc) and were extracted twice. For C7+*Mtb*, a brine solution was additionally used to facilitate the separation of organic layer

from the aqueous layer. The organic layer obtained following ethyl acetate extraction was evaporated to dryness at 30°C in a rotary evaporator. The total EtOAc extract from F31+*Mtb* and C7+*Mtb* filtrate was confirmed to have anti-*Mtb* activity and fractions were collected manually from LC-MS C-18 column detected by UV at either 254/280 nm for F31+*Mtb* or 334 nm for C7+*Mtb* between 5 min to 12 min of run. Each fraction collected in multiple replicates were pooled and tested for activity against *Mtb* H37Rv. F31+*Mtb* EtOAc extract was taken for further purification of the fraction responsible for anti-*Mtb* activity using prep-TLC in ethyl-acetate:hexane (1:1) solvent system. Flash column chromatography using silica gel and ethyl-acetate:methanol (10:1) solvent system was used to purify active component in C7+*Mtb* extract. The purified fraction with anti-*Mtb* activity was checked for purity on LC-MS and structure identification was done using proton and carbon NMR. The final product was dried evaporated and stored at −80°C.

## Mrx1-roGFP2-mediated redox screen

The Mrx1-roGFP2 reporter plasmid was a kind gift from Dr. Amit Singh [17]. *Mtb* H37Rv expressing Mrx1-roGFP2 reporter construct was grown in 7H9-ADGNTw media containing hygromycin (50 μg/ml). Cultures of $OD_{600} = 0.2$ were treated with patulin, nidulalin A, and auranofin each at 0.5× to 10× MIC while diamide at 0.5 to 10 mM concentrations in 96-well black plates with clear bottom at 37°C. Ratiometric fluorescence at emission 510 nm (Ex: 405 nm/480 nm) was recorded at different time points from 0 h to 48 h. Assays were performed in at least 2 biological replicates.

## Quantification of free thiol pools

*Mtb* H37Rv cultures (0.5 ml) were grown to $OD_{600}$ 0.2 in 7H9-ADGNTw media and were treated with patulin and nidulalin A (0.5×-10× MIC) for 24 h (37°C, 45 rpm). Untreated cultures were used as negative control group while auranofin (0.5× to 10× MIC) and diamide (0.5 mM to 10 mM) were used as additional controls to determine the free thiol levels under the same conditions. Posttreatment, the cells were harvested, washed in phosphate buffer saline, and lysed by bead beating for 2 to 3 times for 45 s at 7,000 rpm in a Roche MagnaLyzer and intermittent cooling on ice for 5 min in thiol detection buffer (100 mM potassium phosphate (pH 7.4), 1 mM EDTA). The supernatants collected were used for free intracellular thiol measurements using a thiol detection kit (Cayman Chemical) as reported previously [40]. Assays were done in triplicates and were representative of 2 biological replicates. Thiol concentration was calculated in nM using a glutathione standard curve and the effect of all the agents is represented as % free intracellular thiol with respect to untreated control taken as 100%. $OD_{600}$ was measured before cell harvesting and the relative fluorescence was normalized to measure the thiol levels in equal number of cells.

## Mycothiol adduct assay

*Mtb* H37Rv cultures were grown to $OD_{600} = 0.8$ to 1.0 in 50 ml 7H9-ADGNTw media. Cells were harvested and washed thrice with phosphate buffer saline by centrifuging at 4,000 rpm for 10 min. Cells were resuspended in 10 ml BSA and detergent free 7H9 media supplemented with glucose (4 g/L) and NaCl (0.8 g/L) and aliquoted in 12-well plates where they were treated with either nidulalin A or patulin at different concentrations (0× to 20×). At the indicated time points (ranging from 0 to 48 h). The cells were killed with 60% acetonitrile and frozen at −80°C. Supernatants obtained after centrifugation at 13,000 g for 10 min were tested for the released mercapturic acid adduct of nidulalin A/patulin indicative of the intracellular mycothiol adducts in *Mtb* by LC-MS on C-18 column in positive and negative ion modes.

### *Rv3054c* promoter-reporter construction

The fluorescent reporter construct for the *Rv3054c* promoter was constructed with an episomal plasmid expressing eGFP under the P*smyc* promoter [41], conferring hygromycin resistance, and replicating in mycobacteria using the pAL5000 replicon. This has previously been described as pOLYG P$_{smyc}$ *eGFP* [16]. We also used the plasmid pGiles p$_{L*}$ mScarlet [16] as the source of DNA encoding the mScarlet fluorescent protein. We PCR amplified pOLYG P$_{smyc}$ *eGFP* using primers oPMF373 and oPMF374 (S1 Table), obtaining a linearized version of the entire plasmid with flanking *PaqCI* restriction sites added via 5' overhangs. The 174-bp region immediately upstream of *Rv3054c* was PCR amplified using primers oPMF369 and oPMF370, adding flanking *PaqCI* sites via 5′ overhangs, and amplified DNA encoding mScarlet using primers oPMF371 and oPMF372 utilizing Q5 DNA polymerase (NEB), and a touchdown PCR protocol [42]. PCR products were gel purified and Golden Gate assembly [43] was performed with *PaqCI* and T4 ligase (NEB). Products of the Golden Gate assembly were transformed into 5-alpha competent *E. coli* cells (NEB) via heat-shock and plated onto LB agar plates supplemented with 200 μg/ml of hygromycin (Sigma). The sequence of the insert (plasmid pPMF62a) was confirmed by whole-plasmid nanopore sequencing (Plasmidsaurus). Plasmid pPMF62a was subsequently transformed into *Mtb* via electroporation [44,45]. Primers used for the construction of pPMF62a are mentioned in S1 Table and the vector map for the reporter plasmid pPMF62a can be seen in S1 Fig. *Mtb* H37Rv expressing pPMF62a-*Rv3054c* promoter vector construct was grown in 50 ml ink-well bottles containing 10 ml 7H9-ADGNTw media supplemented with 50 μg/ml hygromycin. Cultures with the final OD$_{600}$ = 0.2 were treated with different concentrations (0.5× to 10× MIC/MIV) of induced F2+*Mtb*, F31+*Mtb*, C7+*Mtb*, D6+*Mtb*, and D9+*Mtb* fungal filtrates as well as patulin, nidulalin A, and citrinin in 96-well black clear-bottom plates. Untreated/DMSO-treated cells containing reporter plasmid were used as a control group. Diamide and iodoacetamide (0.005 mM to 10 mM) were taken as additional controls. The plates were scanned at 2 different emission wavelengths (Red—Ex.: 488 nm, Em.: 590 nm and Green—Ex.: 488 nm, Em.: 530 nm) at different time points (0 h to 24 h). The ratio of Red/Green fluorescence was used as a measure to determine the *Rv3054c* mediated effect of drugs and fungal filtrates on *Mtb* H37Rv cells. The assay was done with at least 2 biological replicates each from 2 individual experiments.

## Results

### Fungal co-cultivation with *Mtb* induces antitubercular metabolites

To identify differentially expressed fungal metabolites, we screened about 1,500 environmental fungal isolates collected from sphagnum peat bogs grown in the presence or absence of live *Mtb* cells. Culture supernatants were collected and assayed against *Mtb* constitutively expressing an mScarlet fluorescent protein. Hit fungi were selected based upon significant shifts in the supernatant volume required for growth inhibition when the fungi were co-cultivated with *Mtb* (Fig 1A). Varying concentrations of inducer *Mtb* cells were required for optimal inhibitor production dependent on the individual fungal species (Fig 1B). In this screen, we obtained 5 unique fungal isolates displaying induced activity upon co-cultivation with *Mtb*, three of which were *Penicillium* sp. with identical ITS sequences (F2, F50, and F51). Another fungus (C7) was an unrelated *Penicillium* sp. and the final hit (F31) was a *Talaromyces* sp.

To assess the specificity of the induced fungal products for *Mtb*, filtrates from F2, C7, and F31 co-cultures with *Mtb* were counter-screened against a panel of ESKAPE pathogens and the non-pathogenic *Mycobacterium smegmatis* which showed that all 3 filtrates were inactive against the other bacterial species at the concentrations produced in co-culture supernatant

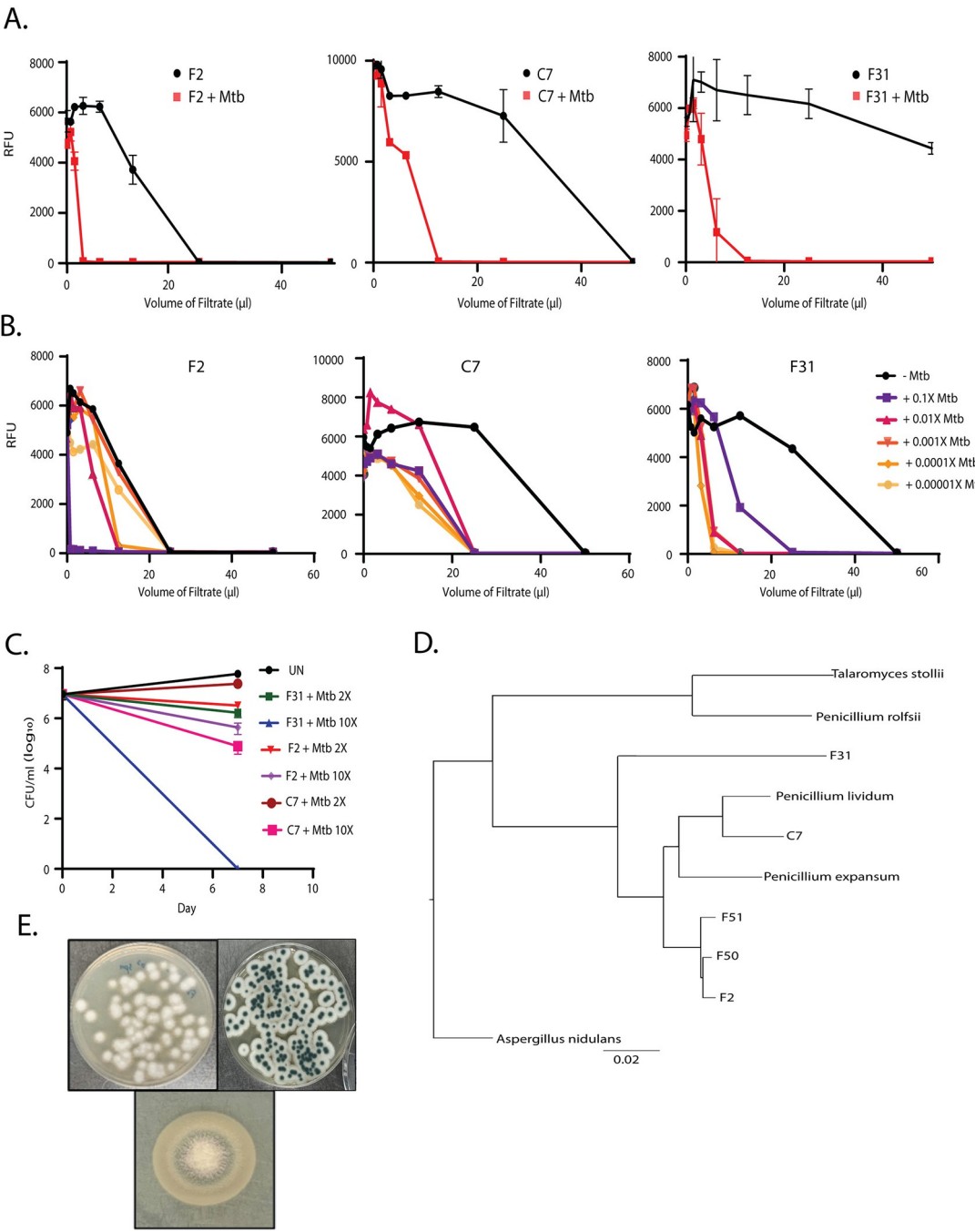

**Fig 1. Activity-based screening of induced fungal hits.** (**A**) Growth Inhibition assays using culture filtrates of 3 hits of mono- and co-culture conditions (*n* = 2–5). (**B**) Dose-dependence of induction with various *Mtb* inocula (X in this figure refers to the density of a stationary phase culture of H73Rv), growth inhibition assays from the co-cultures were done in at least 3 biological replicates. (**C**) Bactericidal activity in terms of decline in log$_{10}$ CFU/ml of co-culture filtrates at 2× and 10× MIV at Day 7. (**D**) Phylogenetic relationship using CVtree [46] alignment between F2, F50, F51; C7 and F31 with their nearest neighbors by their ITS/β-tubulin/calmodulin regions and whole genome sequencing (*P. rolfsii*, *P. expansum*, *P. lividum*, *P. oxalicum*, *P. herquei*, *P. chrysogenum*, and *T. stollii*). *Aspergillus nidulans* was used as an outgroup. (**E**) Morphology of F2 (Left), C7 (Right), and F31 (Bottom) on PDA media plates. Underlying data can be found in S1 Data.

(S2 Table). To rule out the production of siderophores competing for available iron, the induced fungal filtrates were screened for growth inhibition of *Mtb* in the presence of 250 μm Fe$^{+3}$ which revealed that none of the *Mtb* active metabolites were likely siderophores (S3 Fig). The antitubercular metabolite produced by C7 was modestly cidal against *Mtb* reducing CFU by about 1 log after 1 week of exposure, F2 had slightly less cidal activity but the metabolite produced by F31 was highly cidal with no viable *Mtb* detected after 7 days of exposure at this concentration of extract (Fig 1C).

## Determination of fungal reference genomes

Reference genomes for all 5 hits, including the 3 fungi that appeared identical by ITS sequencing (F2, F50, and F51), the *Penicillium* sp. (C7) and the *Talaromyces* sp. (F31), were obtained by whole genome sequencing followed by hybrid assembly. Induced filtrates of F50 and F51 were equipotent against *Mtb* compared to F2 (S2 Fig), although the organisms appeared morphologically unique (Figs 1E and 2E) and all 3 of these fungi were obtained from different geographical locations (S7 Table). The average assembled genome size for all fungal hits was about 33 Mb (~32–35 Mb) and the number of contigs ranged from 61 to 219. Genome data for all the fungal hits is summarized in Table 1. The ITS, β-tubulin, and calmodulin regions of F2, F50, and F51 displayed the closest identity to *Penicillium rolfsii* (>99.5%). Phylogenetic analysis using the alignment-free CVTree also indicated *P. rolfsii* to be the closest neighbor of these 3 fungi (Fig 1D). F31 was most closely related to *Talaromyces stolli*, while C7 appeared more related to *Penicillium lividum*.

## Three related *Penicillium* sp. produce patulin upon co-cultivation with *Mtb*

To identify the biosynthetic gene clusters (BGCs) responsible for anti-*Mtb* activity produced by these fungi, we used RNAseq of co-cultures compared to monocultures for each organism. The 3 related *Penicillium* species (F2, F50, and F51) all showed an induction of a single related BGC upon co-culture with *Mtb* (Fig 2A). In fungus F2, 3 genes of this BGC were among the most highly up-regulated, in F51 9 genes, and in F51 8 genes of this BGC were highly up-regulated compared to monocultured cells. In 2 of 3 of these up-regulated genes included a Type I polyketide synthase. AntiSMASH analysis of this BGC revealed that it shared >78% homology with the patulin producing gene cluster in *P. expansum* (Fig 2A, right bottom and also see S6 Table) [47]. While the patulin gene clusters in F2, F50, and F51 fungal genomes are comparable, the gene organization was found to be distinct from that in *P. expansum* [48]. S6 Table shows the induced genes in F2, F50, and F51 and their amino acid sequence identity with their orthologs in *P. expansum*.

We confirmed the presence of a peak co-eluting with authentic patulin (Fig 2D) in co-culture filtrates of F2+*Mtb*, F50+*Mtb*, and F51+*Mtb*. In each case, this peak was confirmed to be patulin by MS-MS fragmentation of the parent (m/z 155) into 4 product ions (109, 99, 81, 71) by LC-MS/MS (S8 Fig). Patulin was only detected in cultures during co-culture with *Mtb* supporting the notion that synthesis of this metabolite is specifically induced by *Mtb*. Commercially available patulin was confirmed to inhibit growth of *Mtb* with an MIC$_{90}$ of 4.7 μm (S5 Table).

For the *Pencillium* sp. C7, transcriptional analysis of *Mtb*-induced versus monocultures did not result in strong up-regulation of a candidate BGC across biological replicates. Notably, C7 had the least increase in activity upon co-culture where it showed only about a 2-fold increase in potency (Fig 1B). Organic extraction of the co-culture supernatant and activity-guided LC-MS/MS analysis revealed an increased production of the polyketide citrinin in induced filtrates (Fig 2B). The chemical identity of citrinin was confirmed by 2D-NMR (S1 Text).

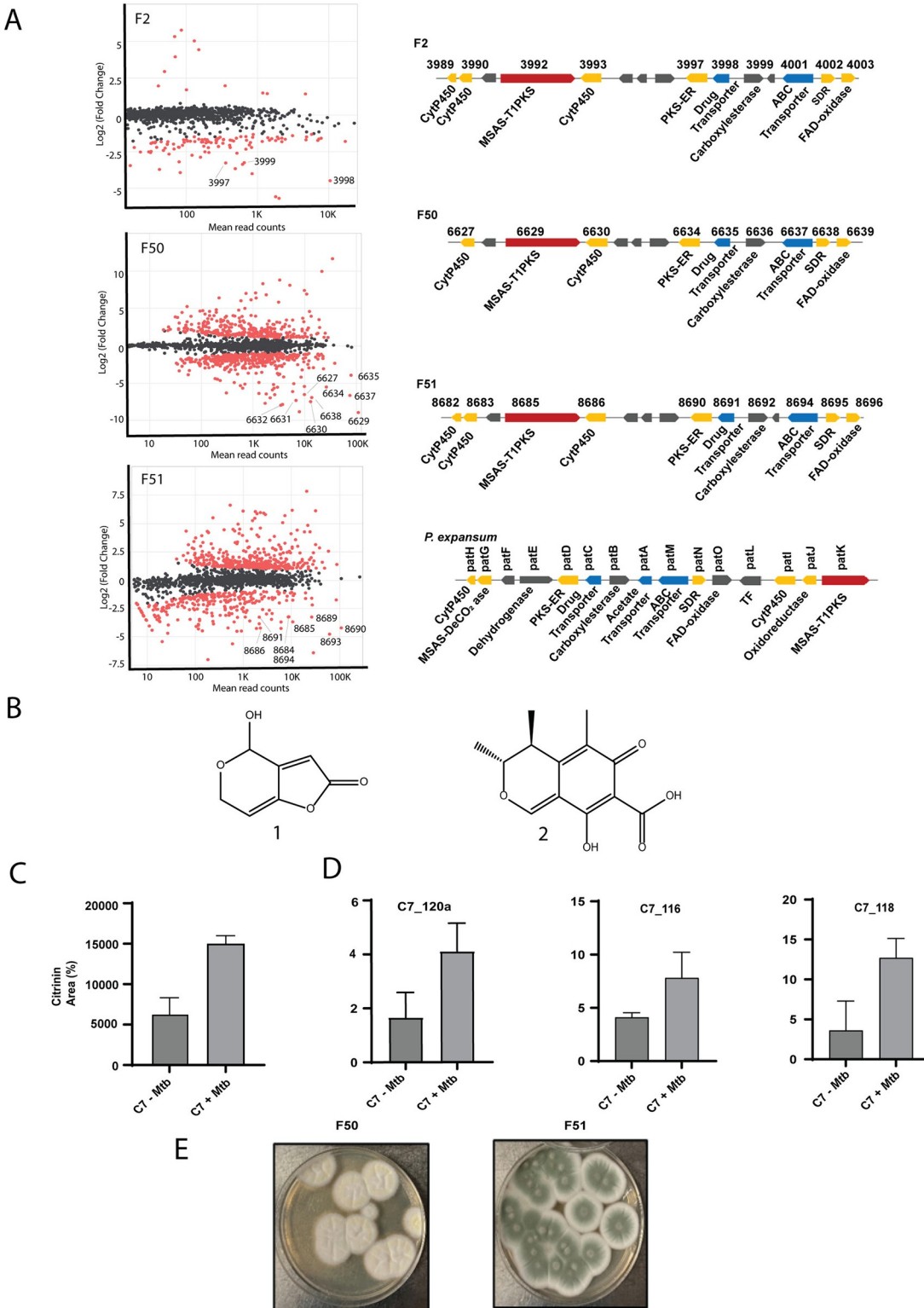

**Fig 2. Four different *Penicillium* sp. produces patulin and citrinin upon co-cultivation with *Mtb*. (A)** Differential gene expression of 3 related *Penicillium* sp. upon co-cultivation with live *Mtb*. Differentially expressed genes along with read counts are shown in MA plots [49]. Up-regulated genes have a negative value in this analysis and red dots represent differentially expressed while gray/black dots are non-differentially expressed genes. The topmost up-regulated BGC for each fungi is shown to the right of each plot and points that are part of this cluster are labeled for each RNAseq experiment. For comparison, the patulin cluster

for *P. expansum* was adapted from elsewhere (bottom right) [48]. (**B**) The chemical structures for patulin and citrinin. (**C**) RT-qPCR of about a 500 bp region of Type I PKS gene in the citrinin BCG (C7_120a) as well as 2 other genes involved in biosynthetic pathway (C7_116 and C7_118) amplified from the total RNA isolated from C7-*Mtb* and C7+*Mtb* conditions. Y-axis is expression fold relative to expression of the ITS region. (**D**) Area percentage of citrinin peaks in mono- and co-culture conditions obtained from the total ion chromatogram (TIC) of C7 -/+ *Mtb* filtrates detected at 334 nm on C-18 liquid chromatography column (Tr = 9.2 min). (**E**) Morphology of F50 and F51 fungal strains on PDA media. Underlying data can be found in S1 Data and S2 Data.

Purified citrinin had an $MIC_{90}$ value of 62 μm against *Mtb* (S5 Table). Despite the apparent lack of differential regulation of the citrinin BGC by RNAseq, qRT-PCR analysis of a portion of a core Type I-PKS gene (C7_120a) as well as 2 other biosynthetic genes for citrinin production (C7_116 and C7_118) indicated 2- to 3-fold higher expression of these genes during co-culture with *Mtb* compared to non-induced cultures supporting the induced biosynthesis of citrinin upon exposure of C7 to *Mtb* (Fig 2C).

## Co-cultivation mediated identification of induced biosynthetic gene clusters in F31 genome

Differential gene expression of the *Talaromyces* sp. (F31) was also performed on monocultures and co-cultures with *Mtb* (Fig 3A). Alignment of the mRNA sequences with the F31 reference genome (>97% read alignment) revealed a potential BGC of 15 clustered genes that were highly overexpressed in co-cultured cells. The genetic organization and proposed gene functions for the 15 genes are shown in Fig 3B and S3 Table (also see S1 Data). BLAST and Interpro Scan analysis of the protein sequences revealed the core enzyme of this BGC as a Type I non-reducing polyketide synthase (Type I NR-PKS; F31_005372). F31_005372 has about 58% sequence identity with the neosartorin biosynthesis enzyme, NsrB [48] containing the characteristic ketoacyl synthase (KS), acyltransferase (AT), and acyl carrier protein (ACP) domains (S4B Fig). Besides these major domains, other functional domains include starter unit ACP transacylase (SAT), product template (PT), phosphopantatheine (PP-ACP), and a dehydrogenase (DH). The PT domain in the NR-PKSs is responsible for catalyzing the specific aldol cyclization and aromatization of mature precursor polyketides. We compared F31_005372 with 55 fungal NR-PKS full length sequences, which have been previously categorized into 8 groups I-VIII [50]. This revealed that F31_005372 is an ortholog of Group V members including XP_657754 and XP_001217072, which are known to produce polyketide synthase derived secondary metabolites like emodin and atrochrysone derivatives (S4A Fig). The PT domain of F31_005372 would therefore most likely result in a C6-C11 cyclization pattern. Domain prediction by InterPro Scan analyses (S4B Fig) indicated the absence of a thioesterase (TE) domain in the protein, which is another characteristic feature of Group V NR-PKS proteins [51]. The protein sequence of F31_005358 shares 74% identity with NsrC that functions as a metallo-β-lactamase thioesterase (MβL-TE) likely responsible for releasing the polyketide precursor (**1**) from the enzyme (S3 Table) [52]. Therefore, based on the sequence identity and homology between most of the proteins encoded by the F31 cluster and their homologues in neosartorin and agnestins pathways [53,54], we propose the following biosynthetic scheme (Fig 3C). Claisen condensation of 7 units of malonyl CoA and 1 acetyl CoA by the F31_005372 would lead to the formation of the cyclized precursor polyketide (**1**), which is released by the MβL-TE (F31_005358) and further decarboxylated by F31_005359 and dehydrated by F31_005364 (which is 58% identical to AgnL8) to generate emodin (**3**). This further undergoes a series of oxidation and reduction reactions by the enzymes that share high sequence identity with those involved in agnestin biosynthesis [53] including an oxidoreductase (F31_005362: 59% identical to AgnL4), short-chain dehydrogenase (F31_005363: 68% identical to AgnL6)

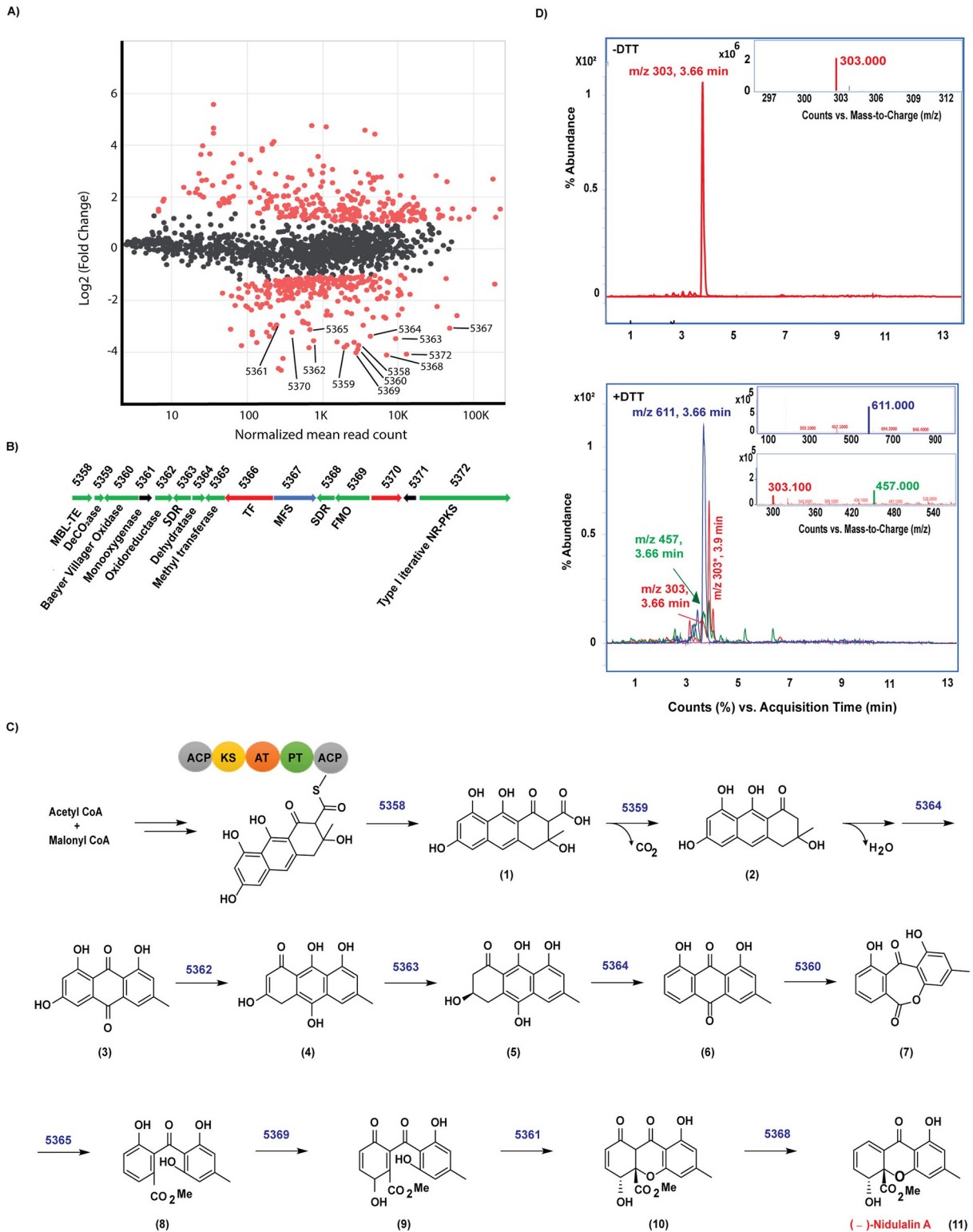

**Fig 3. Proposed biosynthetic pathway for Nidulalin A production.** (**A**) MA plot to represent the $log_2FC$ of genes induced with respect to normalized read counts in co-culture vs. monoculture conditions with genes in the nidulalin A cluster. (**B**) Gene organization of nidulalin A biosynthetic gene cluster in F31. (**C**) Putative biosynthetic pathway for nidulalin A production. Numbers above the arrows in the biosynthetic scheme represents the corresponding F31 genes for the indicated enzymatic reaction. (**D**) Top, chromatogram showing the 303 (m/z) ion detected in -DTT conditions as seen at 3.66 min. Inset, MS plot showing 303 m/z. Bottom, Chromatogram representing 457 and 611 m/z for

the addition of 1 or 2 DTT. Inset: MS plot for 611 m/z ion (top) and 457 m/z ion (bottom). 303* (m/z) is an unrelated ion at Tr 3.9 min. Underlying data can be found in S1 Data and S2 Data.

and dehydratase (F31_005364: 54% identical to AgnL8) to form a dihydroxyanthraquinone (**6**). The Baeyer–Villiger oxidase encoded by F31_005360 shares 60% amino acid identity with NsrF would further oxidize (**6**) to (**7**), which can undergo methylation by a methyl-transferase (F31_005365: 51% identical to NsrG) as well as opening the ring to form (**8**). Oxidation of (**8**) catalyzed by F31_005369 (60% identical to NsrK) encoding an FAD-dependent monooxygenase (FMO) would result in the formation of (**9**). F31_005361 encodes a monooxygenase that shares a weak sequence identity with NsrQ, probably producing (**10**). The only orphan that does not share close homology with any protein in the either the agnestin [53] or neosartorin [54] biosynthetic pathways is a short chain dehydrogenase encoded by F31_005368.

Knowing that we were likely looking for a lipophilic aromatic polyketide, we performed ethyl acetate extraction from F31+*Mtb* culture filtrates followed by fractionation of the extracts on a C-18 liquid chromatography column which revealed that antitubercular activity eluted in a fraction containing an m/z of 303. The isolated peak had growth inhibitory activity against *Mtb* cells with an $MIC_{90}$ of 1.5 μm (S5 Table). Because we suspected this would be a thiol-reactive electrophile, we also examined the reactivity of this metabolite with DTT [39]. Reaction with DTT resulted in the loss of the 303 m/z peak along with the appearance of the 457 Da and 611 Da ions corresponding to addition of 1 (+154 Da) or 2 (+308 Da) DTT molecules (Fig 3D). $^{1}H$ and $^{13}C$ NMR spectroscopy (S1 Text) revealed that the product was a polyketide metabolite previously isolated from *Emericella nidulans* named nidulalin A [55]. As expected, there was minimal detection of nidulalin A in filtrates obtained from monocultures as compared to those from co-cultures as detected by LC-MS/MS, confirming the specific induction of nidulalin A upon co-cultivation with *Mtb* (S5A Fig).

## Fungal metabolites result in a thiol stress response in *Mtb*

Attempts to raise *Mtb* mutants resistant to either patulin or nidulalin A consistently failed to generate high-level resistant mutants. We therefore performed global transcriptional profiling of *Mtb* during exposure to patulin and nidulalin A containing filtrates to identify differentially expressed genes. We excluded citrinin because of its modest activity against *Mtb*. Comparison of the patulin and nidulalin A exposed cells to control *Mtb* cultures revealed a concentration and time-dependent up-regulation of a substantial set of response genes (see S4 Table and S2 Data). Surprisingly, there was a significant overlap between the *Mtb* genes overexpressed in response to either patulin or nidulalin A at similar concentrations relative to their MIC (Figs 4A and S7B). The transcriptional profile exhibited by *Mtb* exposed to the active metabolites from both of these filtrates indicated up-regulation of redox stress pathways, in particular thiol reactive oxidative stress (Figs 4B and S7A) [56,57]. This signature includes a significant dose-dependent expression of genes encoding sigma factors like *sigB*, *sigE*, *sigH*, and their associated regulons which play a role in thiol and redox homeostasis in *Mtb* in response to reactive oxygen and nitrogen species [58]. SigH regulates the expression of SigE, which together control the inducible expression of SigB [59] {Raman, 2001 #19}. The SigH regulon plays a central role in adjusting the thiol-oxidative stress during *Mtb* pathogenesis [56,59–62]. Genes encoding thioredoxin/thioredoxin reductases (*trxC*, *trxB1*, *trxB2*, *thiX*), methyltransferase (*moeZ*), cysteine synthesis/sulfur metabolism/transport (*cysK2*, *cysA1*, *cysH*, *cysM*, *cysN*, *cysO*, *cysT*, *cysW*, *sirA*, *subI*), and lipid synthesis genes (*papA4*) were all found to be overexpressed. There was also an up-regulation of a recently discovered mycothiol-dependent reductase mycoredoxin 2

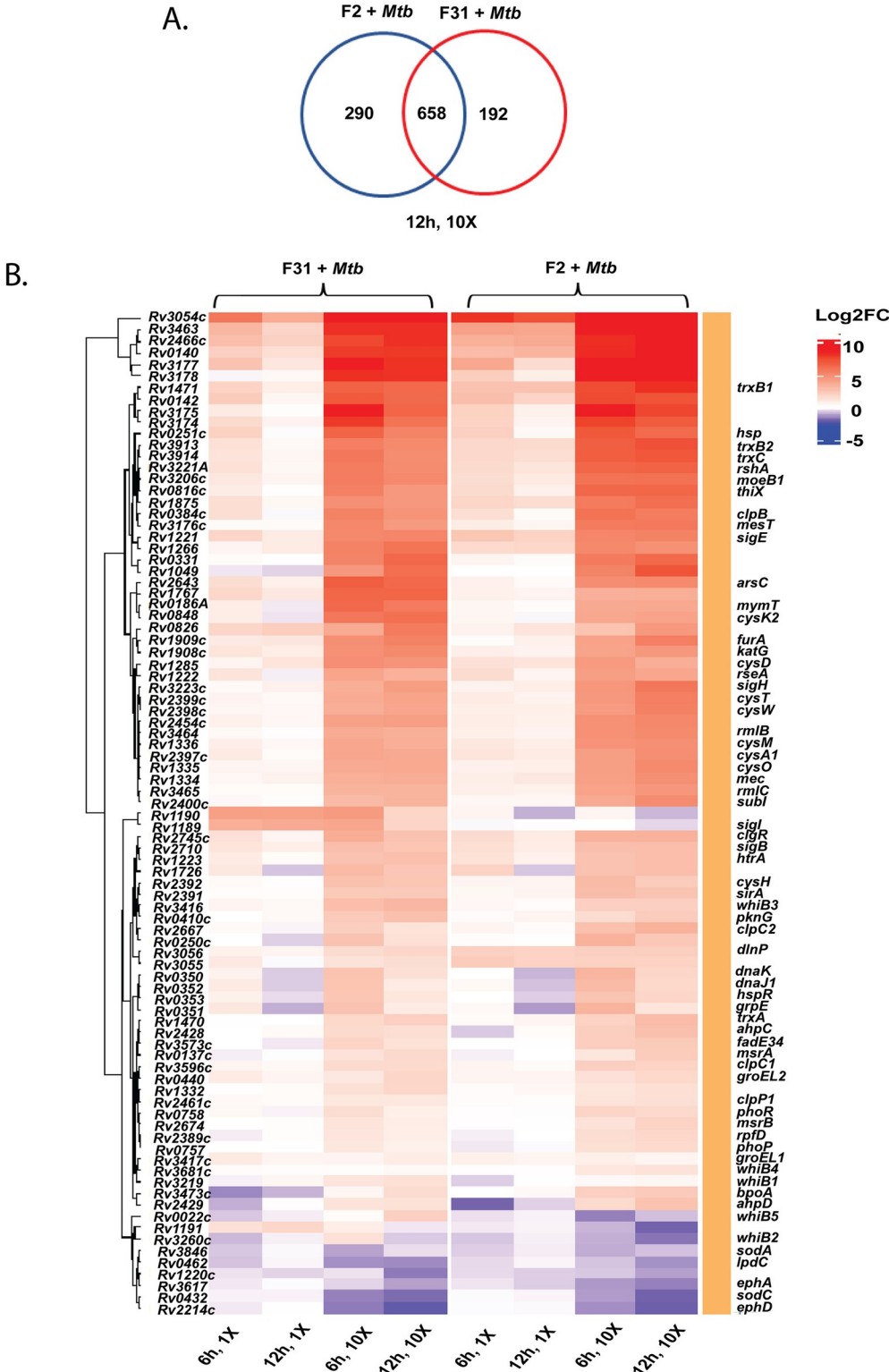

**Fig 4. Thiol stress homeostasis pathways triggered in *Mtb* upon exposure to nidulalin A (F31) and patulin (F2).**
(**A**) Venn diagram representing the overlap of up-regulated genes between F2+*Mtb* and F31+*Mtb* exposed cells at 10×
MIV concentration after 12 h treatment (also see S7B Fig for other treatment conditions). (**B**) Heat map showing the
differential expression of genes involved in redox/thiol stress homeostasis pathways in *Mtb* upon treatment with 1×
and 10× MIV of F31+*Mtb* and F2+*Mtb* culture filtrates for 6 h and 12 h. Differentially expressed genes selected for the

heat plot are statistically significant (FDR <0.1 and $\log_2$FC >1.0) in at least 1 treatment condition. Underlying data can be found in S3 Data.

(*Rv2466c*), several peroxidases, dehydrogenases (*katG, Rv3177, Rv3178, ahpC, ahpD*), ferric uptake transcriptional regulator (*furA*), several probable oxidoreductases (*Rv1726, Rv2454, Rv3463*). In fact, a recent study reported the coordinated expression of SigH and Mrx-2 under oxidative/nitrosative stress is important for the intracellular activation of nitronaphthofuran prodrugs [63]. Moreover, the lack of superoxide dismutase (*sodA, sodC*) and membrane associated oxidoreductases (SseA, DoxX family) up-regulation in our study indicates that the response to the fungal metabolites was not that of a general oxidative stress response [64]. Higher concentrations of the co-culture filtrates similarly resulted in the up-regulation of the thiol stress response with more extensive up-regulation of additional stress responsive genes including several chaperones (*Rv0251, clpB, clgR, grpE, htpx, Rv0991c*), PhoP/R, protein kinase G (*pknG*), phosphoenolpyruvate carboxykinase (*pckA*), several cell envelope lipids or polyketide biosynthesis genes and many members of PE/PPE/PE-PGRS proteins (Figs 4B and S9). The transcriptional profiling data suggests fungal metabolites such as patulin and nidulalin A result in depletion of free intracellular thiols like low molecular weight mycothiol/ergothioneine and reduced cysteines.

## Nidulalin A and patulin deplete the free thiol pool within *Mtb*

Based on the critical role of mycothiol in protecting against oxidative stress within mycobacterial cells, we used the previously reported [65] Mrx1-roGFP2 redox biosensor to measure mycothiol-dependent redox potential during treatment with either patulin or nidulalin A. Cultures were treated with different concentrations of patulin and nidulalin A (0.5× to 10× MIC$_{90}$) for 0–48 h and responses compared to untreated cells or cells treated with auranofin (Aur; 0.5× to 10× MIC) and diamide (0.5 mM to 10 mM). Auranofin disrupts thiol-redox balance by inhibiting bacterial thioredoxin reductase [66,67], whereas diamide causes reversible oxidation of intracellular thiols [56,68,69]. Both patulin and nidulalin A treatment led to a 2- to 4-fold increase in Mrx1-roGFP2 fluorescence ratio at 5× and 10× concentrations at 48 h, consistent with increased mycothiol oxidation (Fig 5A). As expected, auranofin and diamide similarly resulted in an oxidative shift (approximately 3- to 5-fold) (Fig 5A).

We also measured free thiol pools after exposure of *Mtb* cells to either patulin or nidulalin A at 0.5–1× MIC and observed a 25%–50% depletion with complete depletion at 5×–10× MICs compared to the untreated control. Free thiol depletion during patulin or nidulalin A treatment was higher than that observed for auranofin but comparable to those obtained during diamide treatment (Fig 5B). Citrinin is also predicted to be a Michael acceptor which was supported by the observation that this metabolite and the corresponding C7 +*Mtb* fungal filtrate also resulted in depletion of cellular thiols (S6A Fig). Subsequently, 2 other co-culture derived fungal cell-free supernatants (D6 and D9) identified while this work was in progress were also found to result in depletion of intracellular thiols in *Mtb* (S6B Fig). We confirmed that nidulalin A formed mycothiol adducts in vivo within *Mtb* as measured by the release of a nidulalin A-mercapturic acid covalent conjugate in the extracellular milieu [70,71] (S5B Fig).

The most highly up-regulated gene in response to thiol stress was *Rv3054c*. This gene has previously been reported to be up-regulated by diamide and regulated by SigH [57,72]. We therefore created a reporter plasmid where the *Rv3054c* promoter drives expression of the mScarlet fluorescent protein with expression of the eGFP fluorescent protein under a constitutive promoter. *Mtb* carrying this reporter construct demonstrated an increased red to green

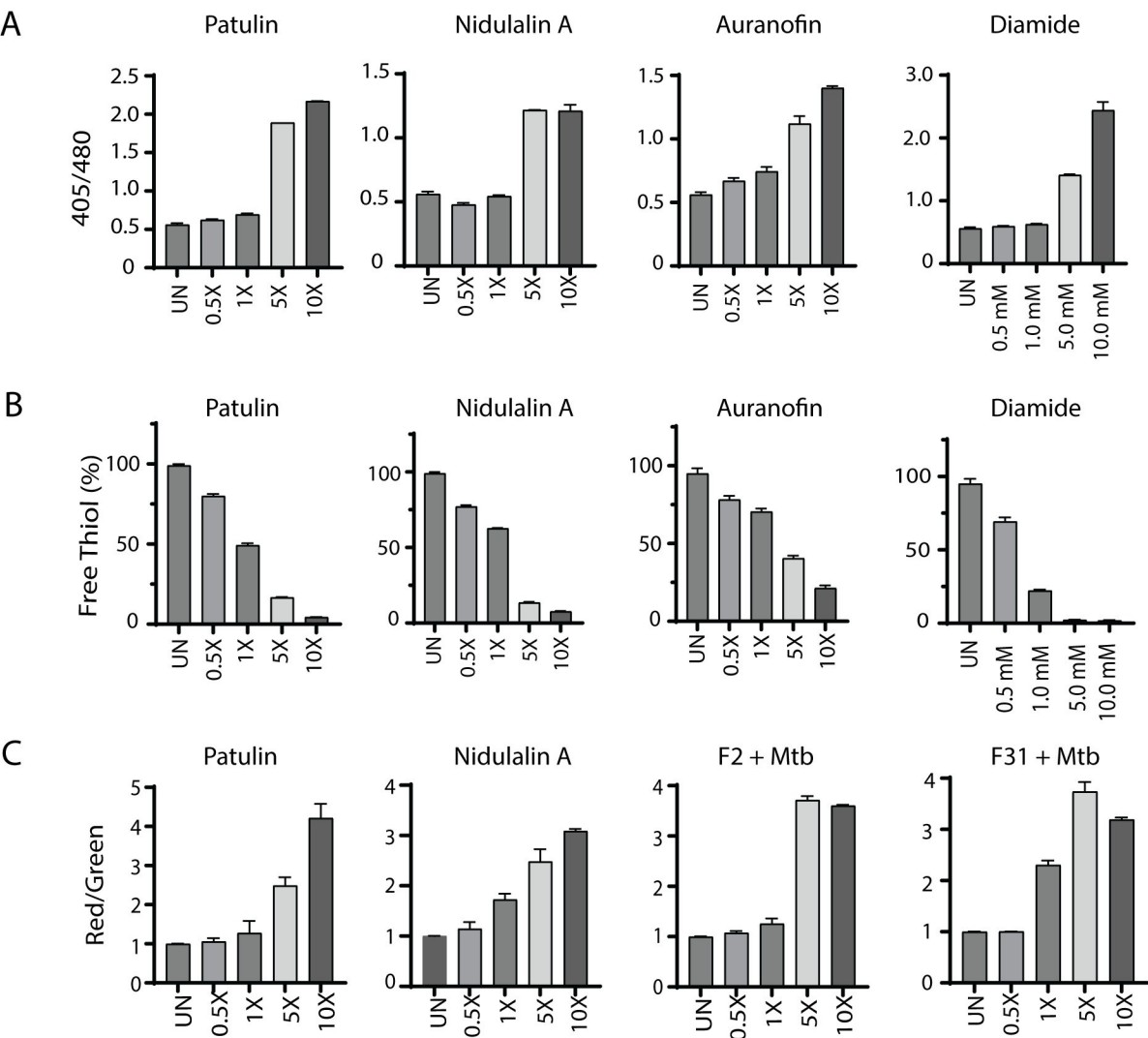

**Fig 5. Patulin and nidulalin A induce a thiol reactive oxidative shift within *Mtb*.** (**A**) *Mtb* H37Rv cells expressing Mrx1-roGFP2 grown to $OD_{600}$ 0.1 were treated with patulin and nidulalin A at different concentrations for 48 h. Diamide, auranofin were used as positive controls for inducing thiol stress. Ratiometric sensor response (Ex:405 nm/488 nm; Em:510 nm). (**B**) Quantitation of free thiol pool was evaluated in *Mtb* H37Rv cells treated with patulin and nidulalin A at different concentrations for 24 h. Free thiol levels were detected by measuring the relative fluorescence at Em: 510 nm (Ex: 380 nm) using a fluorescent thiol detection assay. Relative fluorescence was normalized for the $OD_{600}$ for all the samples tested with respect to the control group. A glutathione (GSH) standard curve was used to estimate free thiol levels. Auranofin and diamide were used as thiol stress controls. The assays results presented here were performed in at least 2 biological replicates. (**C**) *Rv3054c* promoter driven expression of mScarlet normalized to constitutive eGFP in the same cell. UN represents the untreated control cultures. Underlying data can be found in S1 Data.

fluorescence ratio in response to a concentration range of nidulalin A, patulin, their corresponding induced culture filtrates (F2+*Mtb*, F31+*Mtb*), diamide, and iodoacetamide (Figs 5C and S6C). Higher concentrations of iodoacetamide resulted in decreased mScarlet fluorescence likely due to a corresponding decline in *Mtb* viability (S6Cii Fig). To demonstrate the global relevance of thiol stress as an anti-tubercular mechanism, the *Rv3054c* promoter-mediated increase in fluorescence was also confirmed for other induced fungal filtrates (C7+*Mtb*, D6 +*Mtb*, D9+*Mtb*) (S6D Fig) identified in our initial screens.

## Discussion

Co-cultivation is increasingly being used to unlock cryptic natural products that are not normally expressed under standard in vitro growth conditions [73]. Mycobacteria, like most members of the Mycobacteriales order, have unique and complex cell surfaces that offer potential for recognition by competitors. In fact, mycolic acid producing bacteria have been shown to induce the expression of silent secondary metabolites in other actinomycetes [74]. Our results support the use of mycobacteria as inducers of secondary metabolites in fungi.

Both fungi and mycobacteria are found in the gray decomposition layer of the sphagnum peat bog where they compete for limited nutrients as the cell walls of sphagnum moss are slowly decomposed. In this microenvironment, which appears to support the highest concentration of mycobacteria found anywhere in nature thus far, secondary metabolite production by fungi to enhance their ability to compete makes sense. In our initial screen of the ~1,500 fungi isolated from sphagnum samples at or near the gray layer cores, we identified a total of 5 fungi that showed robust induction of antimycobacterial activity upon exposure to live cells of *Mtb*. Three of these fungi were closely related with identical ITS sequences but were isolated from different geographical areas and upon whole genome sequencing were found to be distinct yet closely related. These 3 fungi induced expression of patulin, a known mycotoxin, in response to the presence of *Mtb*. Patulin synthesis in *P. expansum* is controlled by the transcription factor PatL in response to a wide variety of poorly characterized environmental features including carbon catabolism, light, and pH [75]. The fourth *Penicillium* sp. we identified by co-culture weakly induced another complex mycotoxin, citrinin. Like patulin, regulation of citrinin production in fungi is complex and multifactorial.

The fifth fungus induced the expression of a dihydroxyxanthone metabolite that has been previously described from a *Penicillium* sp. from a Japanese soil sample called nidulalin A [76]. Initially identified as having antitumour activity nidulalin A and derivatives have been explored as Topoisomerase II inhibitors and have been associated with a range of bioactivities [77–79]. Several total syntheses of nidulalin A have been reported [80–82] but no further details of its biosynthesis have been reported previously. Our results determined the biosynthetic origins of nidulalin A. Nidulalin A biosynthesis shares many steps with the biosynthetic pathways to agnestin [53] and neosartorin [54], distinguished by a unique short-chain dehydrogenase that catalyzes the penultimate step.

Remarkably, the 3 metabolites produced by these 5 unique fungi all target the same physiological process in *Mtb* cells: thiol homeostasis. This suggests that fungi have selected for the most vulnerable target in mycobacteria under the specific microenvironmental conditions of the gray-layer of a sphagnum bog and identified thiol stress as being the most effective target. These results contrast with the results of a recent genome-wide CRISPR interference screen assessment of target vulnerability [83], but this is not surprising as that screen was done under conditions of aerobically replicating cells. Although mycotoxins, per se, are not promising starting points for drug discovery, they do point to a rich target area that could be exploited and there are many other strategies for drug discovery that could lead to enhanced thiol oxidative stress. Recent screening that has been done under acidic conditions, which amplifies the sensitivity towards imbalances in redox poise, has identified active compounds that perturb thiol homeostasis and give a transcriptional response very similar to that reported here [84,85]. Inhibitors of IspQ, a putative redox sensor, gave rise to compounds that induced thiol stress [86]. Screening of thiol-deficient strains to look for differential activity of hits has led to the identification of compounds that can modulate thiol and oxidative stress levels [87].

Overall, our results suggest that to eradicate the most difficult to kill *Mtb* that exist within hypoxic, caseous lesions and cavity walls, compounds that target thiol homeostasis have the

potential to have significant impact. Improving the clearance of these bacilli has a high likelihood of reducing the overall treatment duration for TB patients.

## Supporting information

**S1 Data. Underlying data for quantitative summary figures in main and supplemental figures.**
(XLSX)

**S2 Data. Fungal RNAseq datasets.**
(XLSX)

**S3 Data. *Mtb* RNAseq data.**
(XLSX)

**S1 Text. Supplemental chemistry section.**
(DOCX)

**S1 Table. Primers used in the study.**
(DOCX)

**S2 Table. MIVs (µL) for F2+Mtb, C7+Mtb and F31+Mtb filtrates against different bacteria.**
(DOCX)

**S3 Table. Differential expression of F31 gene cluster in co-culture conditions and their amino acid identity with proteins belonging to agnestin or neosartorin pathway.**
(DOCX)

**S4 Table. Differential expression of Mtb genes in response to F2+Mtb and F31+Mtb filtrates.**
(DOCX)

**S5 Table. MICs (µM) for nidulalin A, patulin and citrinin against Mtb H37Rv in different media.**
(DOCX)

**S6 Table. Differential expression of patulin cluster in F2, F50 and F51 upon co-cultivation and their sequence identity with their homologs in P.expansum.**
(DOCX)

**S7 Table. Source Information for fungal hits identified in the study.**
(DOCX)

**S1 Fig. Vector map for the reporter plasmid pPMF62a.**
(DOCX)

**S2 Fig. Growth inhibition activity assays in different media.**
(DOCX)

**S3 Fig. No siderophore activity in fungal filtrates.**
(DOCX)

**S4 Fig. Phylogenetic and protein characterization of F31_005372.**
(DOCX)

**S5 Fig. Nidulalin A detection in cell-free supernatant.**
(DOCX)

**S6 Fig. Thiol-reactive oxidative stress in Mtb H37Rv in response to induced fungal products.**
(DOCX)

**S7 Fig. Thiol stress response in Mtb.**
(DOCX)

**S8 Fig. Patulin scanning in induced fungal filtrates.**
(DOCX)

**S9 Fig. Heat map showing differential expression of genes involved in variety of pathways within Mtb upon treatment with F31+Mtb and F2+Mtb filtrate.**
(DOCX)

## Acknowledgments

We thank Dr. Amir Sayed Mousavi and Dr. Pavel Khil (Department of Laboratory Medicine, Clinical Center, NIH) for help with fungal identification. We would also like to thank Dr. Adrian Zelazny (Department of Laboratory Medicine, Clinical Center, NIH) for the ESKAPE pathogens and Dr. Amit Singh (International Centre for Genetic Engineering and Biotechnology, New Delhi, India) for the Mrx-roGFP2 construct. We would like to thank Dr. Robert O'Conner (Laboratory of Bioorganic Chemistry, NIDDK/NIH) for assistance with the 2D NMR spectra. We would also like to thank Rosanna Springston and Gauley District Ranger Richard Raione (US Forest Service) for guiding sampling at Cranberry Glades.

## Author Contributions

**Conceptualization:** Neha Malhotra, Peter Finin, Jessica Medrano, Jenna Andrews, Helena I. M. Boshoff, Clifton Earl Barry, III.

**Data curation:** Sangmi Oh, Justin Lack.

**Formal analysis:** Helena I. M. Boshoff, Clifton Earl Barry, III.

**Funding acquisition:** Clifton Earl Barry, III.

**Investigation:** Neha Malhotra, Sangmi Oh, Peter Finin, Jessica Medrano, Jenna Andrews, Michael Goodwin, Tovah E. Markowitz, Justin Lack, Helena I. M. Boshoff, Clifton Earl Barry, III.

**Methodology:** Neha Malhotra, Sangmi Oh, Peter Finin, Jessica Medrano, Jenna Andrews, Michael Goodwin, Tovah E. Markowitz, Justin Lack, Helena I. M. Boshoff.

**Project administration:** Helena I. M. Boshoff, Clifton Earl Barry, III.

**Resources:** Clifton Earl Barry, III.

**Software:** Tovah E. Markowitz, Justin Lack.

**Supervision:** Helena I. M. Boshoff, Clifton Earl Barry, III.

**Writing – original draft:** Helena I. M. Boshoff, Clifton Earl Barry, III.

**Writing – review & editing:** Neha Malhotra, Sangmi Oh, Peter Finin, Jessica Medrano, Jenna Andrews, Michael Goodwin, Helena I. M. Boshoff, Clifton Earl Barry, III.

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
