## [Editor Report · Decision Letter 0]

19 Sep 2024

Dear Dr Barry, III, 

Thank you for submitting your manuscript entitled "Environmental sphagnum-associated fungi target thiol homeostasis to compete with Mycobacterium tuberculosis." for consideration as a Research Article by PLOS Biology.

Your manuscript has now been evaluated by the PLOS Biology editorial staff, as well as by an academic editor with relevant expertise, and I am writing to let you know that we would like to send your submission out for external peer review.

Once your full submission is complete, your paper will undergo a series of checks in preparation for peer review. After your manuscript has passed the checks it will be sent out for review. To provide the metadata for your submission, please Login to Editorial Manager (https://www.editorialmanager.com/pbiology) within two working days, i.e. by Sep 21 2024 11:59PM.

Kind regards,

Melissa

Melissa Vazquez Hernandez, Ph.D.

Associate Editor

PLOS Biology

---

## [Decision Letter · Decision Letter 1]

22 Oct 2024

Dear Dr Barry, III,

Thank you for your patience while your manuscript "Environmental sphagnum-associated fungi target thiol homeostasis to compete with Mycobacterium tuberculosis." was peer-reviewed at PLOS Biology. Please note that I am currently handling your manuscript since my colleague Melissa is away from the office this week. Your manuscript has now been evaluated by the PLOS Biology editors, an Academic Editor with relevant expertise, and by two independent reviewers. 

As you will see, both reviewers are positive about your study and think it is well-done and important for the field. Based on the reviews, I am pleased to say that we are likely to accept this manuscript for publication, provided you satisfactorily address the remaining points raised by Reviewer #2 to enhance the clarity of the manuscript. In addition, please also make sure to address the following data and other policy-related requests that I have provided below (A-E):

(A) We routinely suggest changes to titles to ensure maximum accessibility for a broad, non-specialist readership. In this case, we would suggest a minor edit to the title, as follows. Please ensure you change both the manuscript file and the online submission system, as they need to match for final acceptance:

“Environmental fungi target thiol homeostasis to compete with Mycobacterium tuberculosis"

(B) You may be aware of the PLOS Data Policy, which requires that all data be made available without restriction: http://journals.plos.org/plosbiology/s/data-availability. For more information, please also see this editorial: http://dx.doi.org/10.1371/journal.pbio.1001797

-Supplementary files (e.g., excel). Please ensure that all data files are uploaded as 'Supporting Information' and are invariably referred to (in the manuscript, figure legends, and the Description field when uploading your files) using the following format verbatim: S1 Data, S2 Data, etc. Multiple panels of a single or even several figures can be included as multiple sheets in one excel file that is saved using exactly the following convention: S1_Data.xlsx (using an underscore).

-Deposition in a publicly available repository. Please also provide the accession code or a reviewer link so that we may view your data before publication. 

Figure 1A-D, 2A-C, 3A, 4A-B, 5A-C, S2A-F, S3, S4A-B, S5A-B, S6A-D, S7A-B, S8, S9

(C) Please also ensure that each of the relevant figure legends in your manuscript include information on *WHERE THE UNDERLYING DATA CAN BE FOUND*, and ensure your supplemental data file/s has a legend.

(D) Per journal policy, if you have generated any custom code during the course of this investigation, please make it available without restrictions. Please ensure that the code is sufficiently well documented and reusable, and that your Data Statement in the Editorial Manager submission system accurately describes where your code can be found. 

(E) Please note that per journal policy, we do not allow the mention of "data not shown", "personal communication", "manuscript in preparation" or other references to data that is not publicly available or contained within this manuscript. Please either remove mention of these data or provide figures presenting the results and the data underlying the figure(s).

We expect to receive your revised manuscript within two weeks. 

*Published Peer Review History*

*Press*

Sincerely,

Richard

Richard Hodge, PhD

rhodge@plos.org

On behalf of:

Melissa Vazquez Hernandez, Ph.D.

Associate Editor, PLOS Biology

Reviewer remarks:

Reviewer #1: The article entitled "Environmental sphagnum-associated fungi target thiol homeostasis to compete with Mycobacterium tuberculosis." by Malhotra et al. represents an ellegant, extensive and complete example of how co-culture-dependent natural products directed at anti-tubercular compound identification can be approached. The authors are praised by the use of multiple complementary techniques such as genomics, bioinformatics, natural product purification and transcriptomics to not only identify potential anti-tubercular agents but also understand their mechanisms of action.

The fact that patulin, citrinin and nidulalin A deplete thiols is of great interest, as they are evolutionary and chemically distinct. Currently, no drugs working by such a mechanism exist, and therefore, the authors have described and validated a new mechanism of action for potential novel antibiotics.

The discovery of the biosynthetic gene cluster responsible for the biosynthesis of nidulalin A is a nice bonus.

Overall, I believe the results disclosed in the manuscript to be novel and of interest to researchers working on mycobacteriology and antitubercular drug discovery, and also of others looking at identifying antibacterial agents.

I have no critiques at this stage.

Reviewer #2: The manuscript by Barry et al reports a very nice approach for finding vulnerabilities that competitors of mycobacteria have 'selected' to compete with and combat these bacteria. The authors screen ~1500 fungi, find three that effectively reduce Mtb RFUs, and subsequently link these to three natural products, all of which recapitulate the co-culture phenotype. Patulin and nidulanin are especially potent. Remarkably, all are electrophilic thiol acceptors and subsequent studies show that they generate thiol stress in Mtb, probably by trapping mycothiol, which has been known to be important for Mtb proliferation. 

Overall, this is high-quality work and the manuscript is very well-written. It can be published more-or-less as is. I only provide a few comments to perhaps further enhance/clarify the work:

- fungus (singular) vs fungi (plural) - please ensure these are used correctly throughout the manuscript.

- it is quite remarkable that only a single biosynthetic locus was induced with fungus F2. Just out of interest, what are some of the non-biosynthetic genes that are upregulated and is there any theme among the three strains here? 

- It would be nice to add the structure of the mycothiol-trapped nidulanin (Fig. S5). Even if the authors do not have conclusive NMR evidence, a proposed structure could be added. The authors may also consider moving parts or all of Fig. S5 to the main text, as this is a key figure. Just a suggestion, ultimately at the discretion of the authors. 

- "In this microenvironment, which appears to support the highest concentration of mycobacteria found anywhere in nature..." I think "known" or "thus far" needs to be added somewhere in this sentence, as clearly all natural environments/microenvironments have not been investigated.

- do the authors have any clues regarding the nature of the elicitor (from Mtb) for these three fungal products?

---

## [Editor Report · Decision Letter 2]

27 Oct 2024

Dear Dr Barry, III,

Thank you for the submission of your revised Research Article "Environmental fungi target thiol homeostasis to compete with Mycobacterium tuberculosis." for publication in PLOS Biology. On behalf of my colleagues and the Academic Editor, Max Gutierrez, I am pleased to say that we can in principle accept your manuscript for publication, provided you address any remaining formatting and reporting issues. These will be detailed in an email you should receive within 2-3 business days from our colleagues in the journal operations team; no action is required from you until then. Please note that we will not be able to formally accept your manuscript and schedule it for publication until you have completed any requested changes.

PRESS

Sincerely, 

Melissa

Melissa Vazquez Hernandez, Ph.D., Ph.D.

Associate Editor

PLOS Biology
